# A primordial noble gas component discovered in the Ryugu asteroid and its implications

Alexander B. Verchovsky [1] ✉, Feargus A. J. Abernethy [1], Mahesh Anand [1], Ian A. Franchi [1], Monica M. Grady [1], Richard C. Greenwood [1], Simeon J. Barber[1], Martin Suttle [1], Motoo Ito [2,3], Naotaka Tomioka [2], Masayuki Uesugi [4], Akira Yamaguchi [5], Makoto Kimura [6], Naoya Imae [7], Naoki Shirai [8], Takuji Ohigashi [7,8], Ming-Chang Liu[9], Kentaro Uesugi [4], Aiko Nakato [5,10], Kasumi Yogata [10], Hayato Yuzawa [7], Yuzuru Karouji [11], Satoru Nakazawa [10], Tatsuaki Okada[10], Takanao Saiki[10], Satoshi Tanaka[10], Fuyuto Terui[10], Makoto Yoshikawa [10], Akiko Miyazaki[10], Masahiro Nishimura[10], Toru Yada[10], Masanao Abe[10], Tomohiro Usui [10], Sen-ichiro Watanabe [12], Yuichi Tsuda[10,13] & Consortium Phase2 curation team Kochi*

Ryugu is the C-type asteroid from which material was brought to Earth by the Hayabusa2 mission. A number of individual grains and fine-grained samples analysed so far for noble gases have indicated that solar wind and planetary (known as P1) noble gases are present in Ryugu samples with concentrations higher than those observed in CIs, suggesting the former to be more primitive compared to the latter. Here we present results of analyses of three fine-grained samples from Ryugu, in one of which Xe concentration is an order of magnitude higher than determined so far in other samples from Ryugu. Isotopically, this Xe resembles P1, but with a much stronger isotopic fractionation relative to solar wind and significantly lower $^{36}Ar/^{132}Xe$ ratio than in P1. This previously unknown primordial noble gas component (here termed P7) provides clues to constrain how the solar composition was fractionated to form the planetary components.

"Solar" and "Planetary" are two of the most abundant primordial noble gas components found in solar system materials. The majority of the former is located in the Sun. Solar wind (SW) composition is commonly used as a proxy for the solar component since this is the only physical solar material available for laboratory analysis. Such analyses have been performed on species implanted into the surfaces of airless bodies such as asteroids (via meteorites) or the Moon, as well as into artificial targets exposed to the solar wind in space (Genesis mission[1]),

[1]The Open University, Milton Keynes, UK. [2]Kochi Institute for Core Sample Research, X-star, Japan Agency for Marine-Earth Science and Technology (JAMSTEC), Nankoku, Kochi, Japan. [3]National Graduate Institute for Policy Studies (GRIPS), Nankoku, Kochi, Japan. [4]Japan Synchrotron Radiation Research Institute (JASRI/SPring-8), Sayo, Hyogo, Japan. [5]National Institute of Polar Research (NIPR), Tachikawa, Tokyo, Japan. [6]Faculty of Science, Kanagawa University, Yokohama, Kanagawa, Japan. [7]UVSOR Synchrotron Facility, Institute for Molecular Science, Okazaki, Aichi, Japan. [8]Institute of Materials Structure Science, High Energy Accelerator Research Organization, Tsukuba, Ibaraki, Japan. [9]Lawrence Livermore National Laboratory, Livermore, CA, USA. [10]Institute of Space and Astronautical Science (ISAS), Japan Aerospace Exploration Agency (JAXA), Sagamihara, Kanagawa, Japan. [11]Core Facility Center, Osaka University, 8-1 Mihogaoka, Ibaraki, Osaka, Japan. [12]Graduate School of Environmental Studies, Nagoya University, Nagoya, Aichi, Japan. [13]The Graduate University for Advanced Studies (SOKENDAI), Hayama, Kanagawa, Japan.*A list of authors and their affiliations appears at the end of the paper. ✉e-mail: sasha.verchovsky@open.ac.uk

or on the surface of the Moon (Solar Wind Composition experiment[2]). SW isotopic and elemental compositions are expected to have been mass fractionated with respect to the true solar composition during the SW formation process. However, the degree of the fractionation remains largely uncertain (see Discussion).

The planetary component (P1) is widespread in solar system materials and is present in many types of meteorites, including carbonaceous and ordinary chondrites[3]. It is located in a carrier phase (Q, standing for quintessence), the nature of which is as yet unidentified, but is known to be very rare, soluble in oxidising acids and mostly carbonaceous. The noble gas concentrations in the Q phase are extremely high[3,4]. Isotopic and elemental compositions of noble gases in this component are different from those in SW. The P1 component (often also called Q) shows some variations in Ne and Xe isotopic compositions in meteorites of different classes and metamorphic grades[4]. The elemental composition of the noble gas component indicates the presence of subcomponents with high $^{36}Ar$[5–7], held in a water-soluble carrier phase. The isotopic composition of Kr and Xe in P1 can be produced from SW by mass fractionation (~10‰ AMU$^{-1}$), though the mass fractionation line does not fit the compositions perfectly. Aside from mass fractionation, the addition of small amounts (1-2%) of r- and s-process isotopes is required to explain the difference between P1 and SW[8–11]. Also, P1 has a significant excess of $^{129}Xe$[8] (when compared with the mass fractionation line), which formed as a result of the decay of extinct $^{129}I$ ($T_{1/2} = 17.6 \times 10^6$ years).

Ryugu is the first C-type asteroid from which material has become available for laboratory study due to the successful Hayabusa2 mission, which collected and returned to Earth a relatively large (compared to the original Hayabusa mission) amount of the asteroid surface matter[12–14]. According to remote spectroscopic data[15], C-type asteroids are built from material very similar or identical to that of carbonaceous chondrite meteorites of the most primitive CI class. Indeed, a number of petrological, mineralogical, geochemical and isotope studies carried out so far[16–22] on the Hayabusa2 samples clearly identify Ryugu as a member of the Ivuna-type (CI) carbonaceous chondrite group, although there are some differences between Ryugu and CI meteorites. One example is in concentrations of the primordial planetary (P1) noble gases[23,24], which are generally higher in the Ryugu material than in CI meteorites. In addition, the Ryugu material is chemically closer to the solar composition than even Orgueil, which is widely used as a proxy for solar composition, and in this sense appears to be more primitive[18], although it has undergone low-temperature (below 400 K) aqueous alteration[19,25].

In the present study, we analysed three fine-grained Ryugu samples (A0219, C0208, and C0209) by high-temperature resolution stepped combustion using our mass spectrometric complex (Finesse instrumentation). He, Ne, Ar, C and N abundances and isotopic compositions were measured in all the samples, while Xe was measured in only two of them. The data for C and N will be discussed elsewhere. Critically, the samples almost entirely avoided exposure to the terrestrial atmosphere prior to analysis. The instantaneous opening of the sample container, which happened as a result of the stress in the metal seal that occurred during the deployment of the parachute in the Earth's atmosphere, did not result in significant contamination because the atmospheric gas pressure inside the container remained ≤ 68 Pa (1000 times less than in the atmosphere), which was determined immediately after the capsule landed[26]. Before and during loading into the Finesse vacuum system for analysis, the samples were kept in an atmosphere of pure nitrogen. For details, see the "Methods" section.

We found that one of the analysed samples (C0209) has a significantly high Xe concentration and an isotopic composition different from any known primordial Xe components.

## Results

### Noble gas concentrations and relative abundance

Data for bulk noble gases and Ne, isotopic compositions in the Ryugu samples and in three CI meteorites, some of which were analysed in the present study, are shown in Table 1. For comparison, the ranges of noble gases, and Ne isotopic composition analysed in Ryugu samples in other studies[23,24] are also shown.

As can be seen from Table 1, $^4He$, $^{20}Ne$, $^{36}Ar$ and $^{132}Xe$ concentrations in the Ryugu samples analysed in the present study and in different laboratories all over the world[23,24], are in the same range except for $^{132}Xe$ in sample C0209 which is an order of magnitude higher than the highest $^{132}Xe$ concentration found in other Ryugu samples. $^4He$, $^{36}Ar$ and $^{20}Ne$ concentrations and Ne isotopic compositions are also similar in all the samples shown in Table 1.

Although concentrations of noble gases in the Hayabusa2 samples are generally higher than in carbonaceous chondrites, the Xe concentration in C0209 is clearly anomalous, so it is necessary to exclude the possibility of any artefacts before it is accepted as real. One possible reason for this observation is contamination from the Xe ion engine used to navigate the Hayabusa2 spacecraft. However, according to the detailed time sequence of the ion engine activity during flight to Ryugu, the last time it operated was about half a year before Hayabusa2 arrived at the asteroid[27]. Even in the very unlikely case of contamination from the engine after that, the Xe isotopic composition must be atmospheric or related to it through mass-dependent fractionation. As can be seen in the next section, Xe in C0209 is not atmospheric and cannot be produced from atmospheric Xe by mass fractionation either (see below and Supplementary Fig. S1). For the same reason, a leak in the vacuum system of the Finesse instrument exclusively related to the Xe analysis can also be excluded. In addition,

**Table 1 | Bulk noble gas concentrations and Ne isotopic composition in the Ryugu and CI samples**

| Sample/ source | $^4$He, cm$^3$ STP/g x10$^{-4}$ | $^{20}$Ne, cm$^3$ STP/g x10$^{-5}$ | $^{20}$Ne/$^{22}$Ne | $^{21}$Ne/$^{22}$Ne | $^{36}$Ar, cm$^3$ STP/g x10$^{-6}$ | $^{132}$Xe, cm$^3$ STP/g x10$^{-8}$ | $^{132}$Xe/$^{36}$Ar |
|---|---|---|---|---|---|---|---|
| **C0208** | 16 | 2.24 | 13.30 ± 0.30 | 0.0335 ± 0.0026 | 3.59 | 1.78 | 0.005 |
| **C0209** | 8.29 | 3.09 | 12.64 ± 0.54 | 0.0386 ± 0.0031 | 2.44 | **40.0** | **0.16** |
| **A0219** | 4.25 | 1.57 | 13.38 ± 0.35 | 0.0350 ± 0.0073 | 3.56 | n.a. | n.a. |
| **Orgueil** | 2.21 | n.a. | n.a. | n.a. | 2.65 | 1.64 | 0.006 |
| **Ivuna** | 0.77 | 0.068 | n.a. | n.a. | 1.67 | 1.68 | 0.010 |
| **Tagish Lake**[37] | 0.62 | 0.41 | 6.23 | 0.204 | 1.27 | 1.58 | 0.012 |
| **Ryugu, Okazaki et al., 2022**[23] | 0.64– 57.1 | 0.0053–1.24 | 6.43–12.87 | 0.0312–0.214 | 1.14–14.2 | 0.37–3.54 | 0.002–0.018 |

Errors for Ne isotope ratios correspond to 1σ. Xe concentration and Xe/Ar ratio in C0209 are shown in bold to distinguish them from other samples.

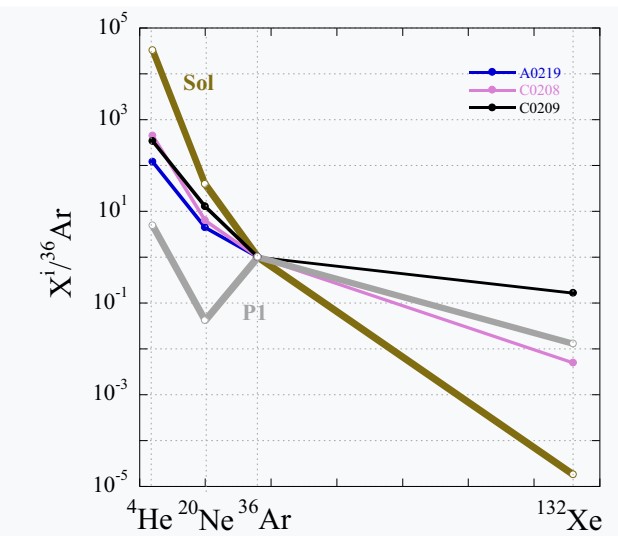

**Fig. 1 | Relative abundance of noble gases in the analysed Ryugu samples.** The elemental ratios are normalised to $^{36}$Ar. Solar wind (SW) and planetary (P1) compositions[3] are shown for reference. The data points for all samples fall between SW and P1 compositions except for the $^{36}$Ar/$^{132}$Xe ratio in C0209. Source data are provided in Table 1.

a very specific release pattern of Xe from C0209 (see below) is also incompatible with such a possibility.

Thus, the high Xe concentration in sample C0209 must be accepted as real and if so, the low $^{36}$Ar/$^{132}$Xe ratio is also real. The relative abundances of measured light noble gases are in between SW and P1 compositions for all the samples, including C0209. The same is true for the Ne isotopic composition in the analysed samples (Table 1 and Supplementary Table S1). For the heavy noble gases, Ar and Xe, the experimental point for C0208 is also in the SW-P1 range, while the sample C0209 is outside of it (Fig. 1).

### Isotopic composition of Xe in C0209
Experimental details for Xe isotope analysis can be found in the "Methods" section, and the He, Ne, Ar and Xe concentrations and Ne and Xe isotopic compositions in the combustion steps are shown in Supplementary Tables S1–S4.

On three-isotope diagrams (except for $^{134}$Xe/$^{132}$Xe vs. $^{136}$Xe/$^{132}$Xe), the data points for combustion steps of the sample C0209 are almost all within the triangle with Xe-P1, Xe-HL (a component of presolar nanodiamonds[3]) and Xe with unknown composition, as end members (Fig. 2). The same remains true if instead of Xe-P1 SW Xe is taken as one of the end-member components. As the Ryugu asteroid represents material similar to CI-type carbonaceous chondrites, it must contain Xe-P1, which is confirmed by laboratory studies[23,24] of the Ryugu samples. On the other hand, the Ryugu surface regolith material collected by the Hayabusa2 mission was definitely exposed to SW radiation and, therefore, must contain a SW component. This is also confirmed by analyses of the Ryugu samples[23,24], however in most samples Xe-P1 dominates over SW. Thus, both SW Xe and Xe-P1 can contribute to the Xe isotopic composition of the C0209 sample. The data points corresponding to the main Xe release form a cluster opposite to the P1 or SW Xe end of the mixing lines. At higher temperatures, the data points move towards Xe-P1 (SW Xe) as well as to Xe-HL[3] compositions. The presence of the latter is confirmed by other analyses of the Ryugu samples[23,24] and is due to its release from presolar nanodiamonds, a common component of carbonaceous chondrites[28].

As can be seen on the $^{129}$Xe/$^{132}$Xe vs. $^{136}$Xe/$^{132}$Xe plot (Fig.2), a contribution from $^{244}$Pu fission Xe cannot explain the experimental

data. The same is also true for $^{238}$U fission Xe, which has a $^{136}$Xe/$^{132}$Xe ratio even higher than that for $^{244}$Pu. Contributions from these sources can also be excluded on the basis of low (0.009 ppm[5]) U (and consequently Pu) content in the Ryugu and extremely high Xe concentration in the sample. The required Xe with unknown isotopic composition cannot be produced from air Xe by mass-dependent fractionation either (Fig. 2 and Supplementary Fig. S1).

Thus, Xe with an unknown isotopic composition, which we tentatively named Xe-X just to emphasise that it is an unknown component, is required to explain the observations. The pure composition of this Xe-X can be located on an extension of the mixing lines from P1 (or SW), but where exactly is not known. However, considering Xe isotope variations in combustion steps alongside cumulative Xe yield (Fig. 3), within nearly the entire Xe release, the Xe isotope ratios remain almost constant despite Xe abundance in the steps varying by factors of 2-5. At higher temperatures, after more than 85–95% of Xe is released, the isotope ratios shift to Xe-P1, SW Xe, and Xe-HL compositions in the same way as observed in Fig. 2. In other words, the contribution of Xe-P1 and/or SW Xe to the Xe budget at the temperature steps where Xe isotope ratios remain constant appears to be insignificant, or even absent. Thus, Xe in these low-temperature (100–400 °C) steps can be considered as a pure Xe-X component. This interpretation agrees with the very high Xe concentration in the sample. Xe-P1 is released at a somewhat higher temperature than Xe-X (see below), which is why its signature is seen only after most of the Xe-X is gone. An alternative explanation for the constant Xe isotope ratio in the main Xe release would be that both components (even trapped in comparable amounts) have almost identical release patterns, which seems an unlikely scenario. We calculated the Xe-X isotopic composition as an average for the steps with constant isotope ratios and uncertainties as standard errors of the means (Fig. 3 and Supplementary Table S5).

Thus, Xe-X is a separate primordial Xe component, which has not been observed before in the solar system material.

## Discussion
From the isotope ratios of Xe-X shown in Fig. 3, we calculated isotope ratios normalised to $^{130}$Xe and plotted them in comparison with SW Xe (Fig. 4a).

It is clear that Xe-X can be largely explained by the mass-dependent isotope fractionation of SW Xe. However, the mass-fractionation line does not fit Xe-X perfectly: a significant deviation is observed for $^{129}$Xe and minor (but outside the error bars) for $^{131}$Xe, $^{132}$Xe, $^{134}$Xe and $^{136}$Xe. In this sense, Xe-X is similar to Xe-P1, including deviation at mass 129 and small deviations at other masses from the corresponding mass fractionation lines (Fig. 4b), though the deviations have a different isotope structure for each component and are slightly higher for Xe-X than for Xe-P1. The only significant difference between the components is in the degree of mass fractionation from SW composition: Xe-X is much more strongly fractionated, by a factor of 3.5 than Xe-P1 (Fig. 4a). It is important to note here that Xe-X cannot be produced from Xe-P1 by simple mass fractionation (see Fig. 4c, d and the figure caption).

These observations suggest that Xe-X and Xe-P1 can be considered as two types of planetary components formed from the same solar composition, likely through similar physical processes of mass fractionation, which occurred with different intensities in each case. In other words, Xe-X belongs to the same family of planetary components and can, therefore, be named Xe-P7 (next available number) according to the P nomenclature[28].

The similar (within experimental uncertainties) excess of $^{129}$Xe relative to the fractionation lines observed for both Xe-P1 and Xe-P7 is quite remarkable (Fig. 4b). There is now a general agreement that the excess in Xe-P1 is due to the decay of extinct $^{129}$I. An important

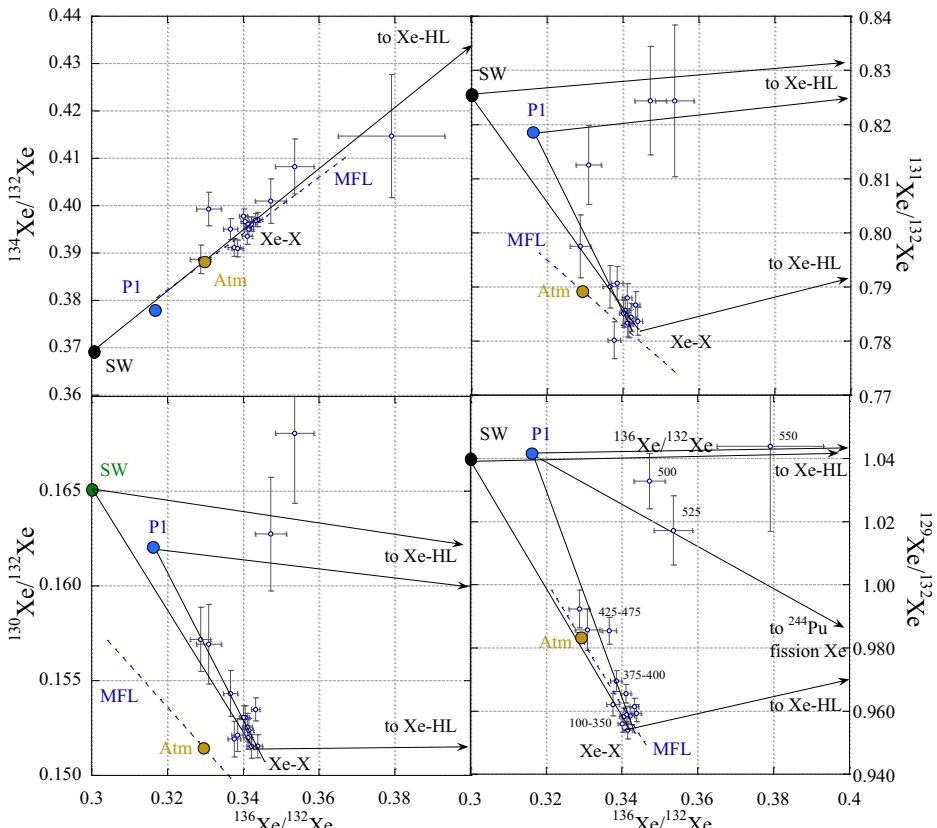

**Fig. 2 | Xe three-isotope diagrams with data points for combustion steps of C0209 (Source data are provided in Supplementary Table S4).** Xe isotopic compositions of the solar wind (SW[11]), atmosphere (Atm.), planetary (P1[4]), and direction to presolar nanodiamonds Xe-HL[28] and to [244]Pu fission Xe[38] (only for the [129]Xe/[132]Xe vs. [136]Xe/[132]Xe plot) are also shown for reference. MFL- mass fractionation line. Most of the experimental points are on the mixing line going through P1, though mixing with SW Xe cannot be excluded. The numbers next to the points on

the [129]Xe/[132]Xe vs. [136]Xe/[132]Xe diagram are the temperature steps in °C. Error bars correspond to 1σ. Dashed lines through atmospheric composition indicate the direction of mass-dependent fractionation. Apart from known primordial noble gas components, an additional unknown component (Xe-X) is required to explain the experimental data. Clearly, Xe-X cannot be produced by fractionation of atmospheric Xe.

observation is that Xe-P1 has the same excess of [129]Xe relative to the underlying mass fractionated SW Xe, and, consequently, the same contribution of radiogenic[129]Xe in meteorites of different classes and petrological types[4,9,29]. There is much less certainty regarding the scenario required to explain the observation. What is clear is that the reservoir from which Xe-P1 was trapped into the Q phase also (apart from noble gases) contained live [129]I, which was trapped into the phase along with Xe. In general, for different trapping mechanisms, it is believed that iodine, as chemically more active than Xe, is trapped more efficiently than Xe. Similar fractionation between I and Xe should be observed even in the case of trapping by ion implantation due to the lower ionisation energy of iodin compared to Xe (10.45 vs. 12.13 eV, respectively). Since accretion of meteorite parent bodies and parent bodies metamorphism occurred before decay of the trapped [129]I was completed, the resulting relative abundance of [129]Xe in Xe-P1 may be the same in different meteorites, if no loss of Xe has occurred in these processes. We consider scenarios in which the loss of Xe from Q during parent body metamorphism occurred simultaneously and in the same proportion for meteorites of different groups and petrological types to be unrealistic. The scenarios considered by J. Gilmour[9] cannot provide this either. The observation that the concentration of Xe-P1 systematically decreases with increasing petrological type of meteorites[29] and [36]Ar/[132]Xe and [84]Kr/[132]Xe ratios in P1 are progressively decreasing with the degree of parent-body alteration[4] do not contradict the suggested scenario with no loss of Xe: the former can only be due to destruction (oxidation) of the carrier itself, and the latter is not

necessarily associated with loss of Xe along with Kr and Ar. The fact that the [36]Ar/[132]Xe ratio does not correlate with [132]Xe concentrations[4] supports the above suggestion. Thus, the same (excess [129]Xe)/[132]Xe ratio in Xe-P1 and Xe-P7 means that both components had the same initial composition, including the [129]I/[132]Xe ratio, before being captured by hosts, emphasising their genetic relationship.

The carrier of Xe-P7 is combustible in a similar manner to the carrier of Xe-P1: both are oxidised in nearly the same relatively low-temperature (100–500 °C) ranges, which are only slightly different from each other (Fig. 5a). Important observation (Fig. 5b) is that release temperature of noble gases are mass dependent: the higher the mass of the species the higher the release temperature except for [132]Xe, which is released before [4]He and before [20]Ne for C0209 and C0208 correspondingly. To explain the observation, it is necessary to consider that in contrast to pyrolysis, in which the release of gases is controlled by diffusion, the combustion release of gases is mostly determined by their spatial distribution within the grains and that combustion generally proceeds on a layer-by-layer basis. Such a specific spatial distribution of noble gases could be a result of their implantation into grains of carriers P1 and P7. Anyway, the observations mean that both carriers appear to be carbonaceous and may be chemically similar. One of the explanations for the very high Xe-P7 concentration in C0209 may be due to its carrier's ability to trap Xe much more efficiently than that of P1 and/or because of the relatively high concentration of P7 carrier itself in the sample.

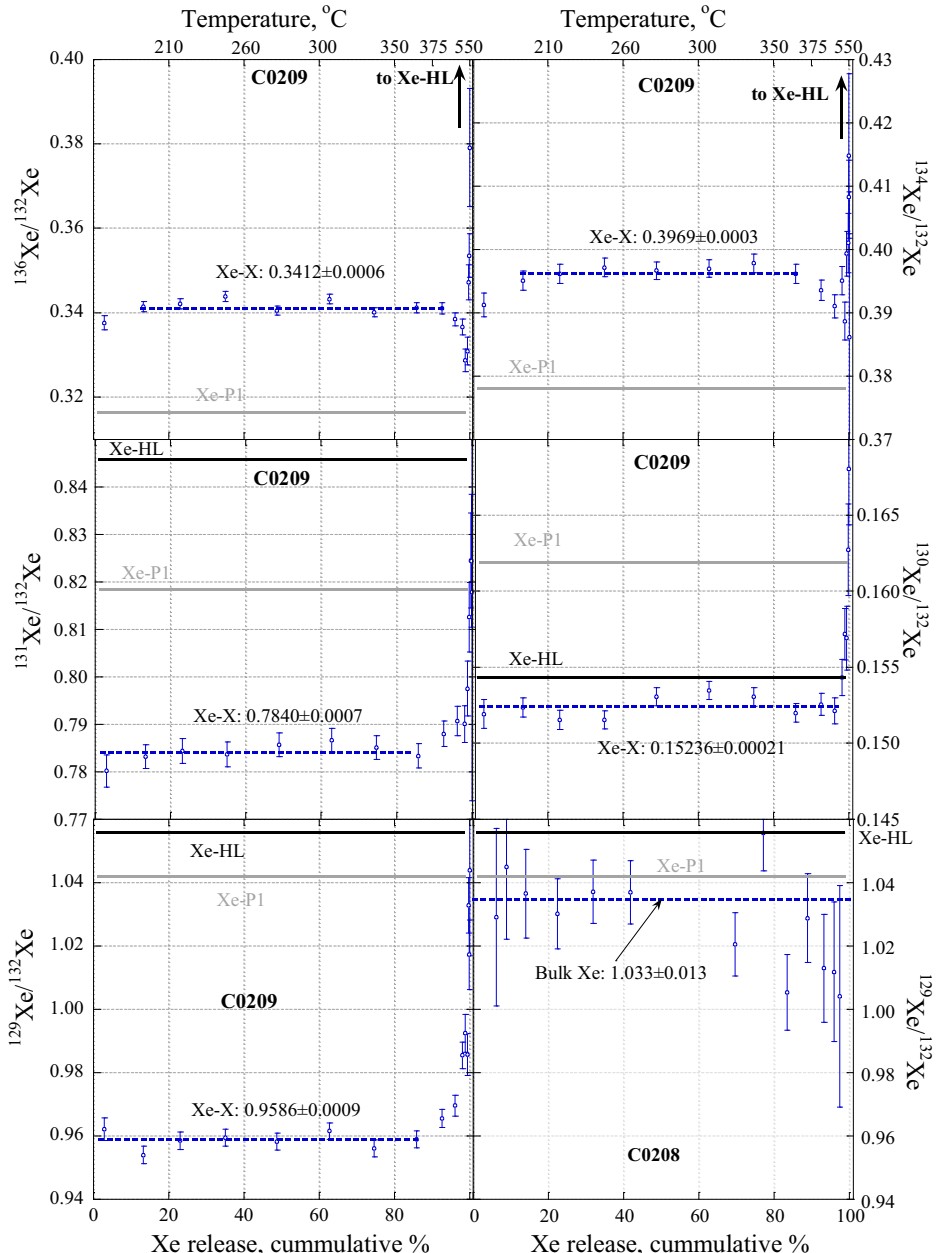

**Fig. 3 | Variation of Xe isotope ratios during stepped combustion of C0209 and C0208 (for $^{129}$Xe/$^{132}$Xe only).** The dashed lines for C0209 define the average ratios calculated for the range of Xe release where the ratios show no significant variations (Source data are provided in Supplementary Table S4). The values for the average ratios are shown next to the lines. The errors are calculated as the standard error of the means. After more than 85–95% of Xe is released, the isotope ratios move towards planetary (Xe-P1[4]) and presolar nanodiamonds Xe-HL[28]. The $^{129}$Xe/$^{132}$Xe ratio variations for C0208 with "normal" Xe concentration are shown for comparison. The dashed line, in this case, corresponds to the bulk value, which is indistinguishable from Xe-P1 within the error bars (1σ).

---

As discussed above, most Xe released from C0209 belongs to the P7 component; the signature of Xe-P1 appears above 400 °C, where most of Xe-P7 is gone (Figs. 3, 4). Since the bulk concentrations and relative abundance of He, Ne and Ar in C0209 are similar to those in C0208, it is reasonable to suggest that these gases in C0209 mostly belong to P1 with some contribution of SW. However, from general considerations, it is impossible to imagine that P7 consists of only Xe without lighter noble gases. It is difficult to estimate how much He, Ne and Ar P7 is present in C0209 but it is clear that the measured $^{36}$Ar/$^{130}$Xe in the sample represents an upper limit for pure P7. Figure 6 shows noble gas elemental ratios for P1 in C0209 normalised to Xe and SW composition. As in Fig. 1 with different normalisation, on this plot, Ar in C0209 is depleted relative to Xe compared to P1 composition.

However, He and Ne are more abundant in C0209 in comparison with P1. This is due to the contribution of the implanted SW component that can clearly be seen in Ne isotopic composition (Fig. 7). It is reasonable to assume that the light noble gases (He and Ne) in P7 should also be depleted with respect to P1, and to a greater extent than Ar. The estimated noble gas element abundance in P7 is shown in Fig. 6 by the dotted line. This estimation assumes that the true Ar/Xe ratio in P7 is an order of magnitude lower than found in C0209 and that He/Ne/Ar ratios in P7 follow the same pattern as observed for P1. Thus, it is likely that due to the very strong elemental fractionation with depletion of light elements relative to heavy ones that occurred during the formation of this component from the solar composition, Xe may be the most abundant element in P7.

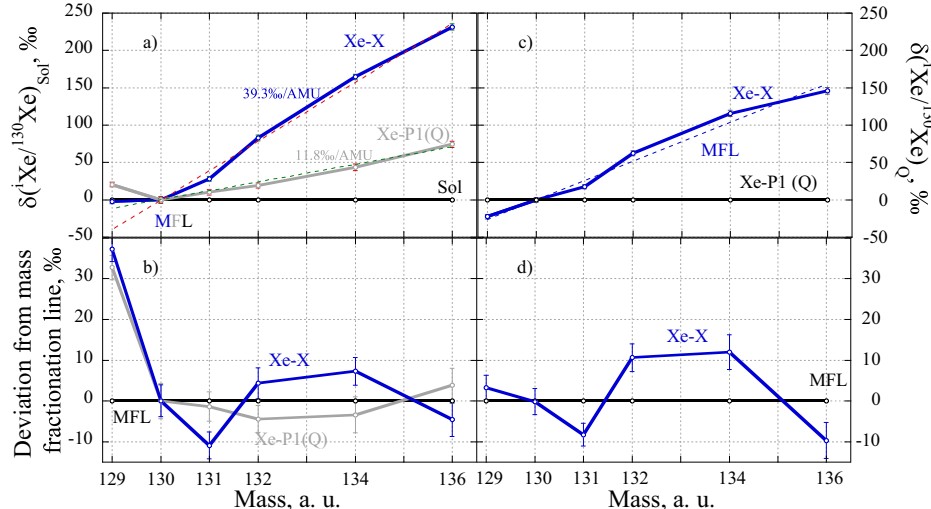

**Fig. 4 | Comparison of Xe-X (Source data are provided in Supplementary Table S5) with Xe-P1 and solar Xe. a** Isotopic composition of Xe-X and Xe-P1 normalised to $^{130}Xe$ and SW Xe composition. Both components are mass fractionated relative to SW Xe with significantly different degrees of fractionation (shown for each line). The mass-fractionation lines (MFL) are the best fits (by the least square method) for the corresponding Xe compositions forced to go through the point (0; 130). **b** deviation of the Xe-P1 and Xe-X from corresponding mass-fractionation lines (solid horizontal line). The deviations are similar for both components at mass 129, which is associated with the decay of extinct $^{129}I$. Apart from mass fractionation, both components require a small addition of r- and s-process isotopes to fractionated SW compositions to fit the observed isotope ratios. As can be seen from (**b**), the additions must be different for Xe-P1 and Xe-X. The chi-squared ($\chi^2$) values determined as defined by Gilmour, 2010[9] (excluding deviation at mass 129) are 3.3 and 18.1 for the Xe-P1 and Xe-X fitting lines, respectively. **c** Isotopic composition of Xe-X normalised to $^{130}Xe$ and Xe-P1(Q). At first glance, Xe-X can be produced from Xe-P1 by mass fractionation. Obviously, this is because both Xe-X and Xe-P1 are closely related to mass fractionation of the same (solar) starting composition. However, in more detail (**d**), deviations from the mass fractionation line are significantly larger than experimental uncertainties ($1\sigma$) at masses 131–136, suggesting that Xe-X cannot be produced from Xe-P1 by simple mass-dependent fractionation. $\chi^2$, in this case, is 32.5.

Obviously, the P7 component is not as widespread as P1: it has never been identified in any carbonaceous or ordinary chondrites and is present here only in one out of about 20 Ryugu samples analysed for Xe so far. The reason for that could be its low-temperature resistance, i.e., it does not survive during thermal parent body metamorphism. Given its primitive status, Ryugu is probably the place where P7 could be preserved. Aqueous alteration does not seem to affect the P7 carrier phase significantly. The fact that this component is not present in all analysed Hayabusa2 samples can be explained by the sample's heterogeneity due to the differences in their thermal history associated with impact events, which may appear on a very local scale. The samples we analysed contained 100–200 grains. If only in one of them Xe-P7 is present then the heterogeneity must be such that, at best, only one grain in a thousand can contain it. In fact, grains containing Xe-P7 should be much rarer, taking into account it was not observed in a great number of primitive meteorites analysed so far. In addition, the difference in the release temperature of Xe-P1 and P7 (Fig. 5) is not so significant to explain the total absence of Xe-P7 in primitive meteorites of CI and CM groups. Therefore, an additional factor must be invoked to explain these observations. One of the possibilities is to suggest that the grain with Xe-P7 identified in the C0209 sample is extraneous to Ryugu asteroid and comes from a parent body, material from which may never have reached Earth in the form of meteorites. In this case, the Xe-P7-bearing grain found in sample C0209 may be the only one present in the material sampled from the Ryugu asteroid by the Hayabusa2 lander. This also explains why this component has not been found in meteorites. Alternatively, if Xe-P7 is a local component, then we believe that sooner or later, it should be identified in other Ryugu samples in any future noble gas studies. One way to achieve this is to analyse a well-homogenised sample obtained from a relatively large aliquot of the Ryugu material. And of cause these studies must include all Xe isotopes to see whether the light Xe isotopes also follow the same fractionation pattern.

On the three-isotope diagram ($^{36}Ar/^{130}Xe$ vs. $^{136}Xe/^{130}Xe$), the compositions of P7, P1 and SW form a trend indicating that element and isotope ratios in the components are positively correlated with each other (Fig. 8), i.e., the higher their enrichment by heavy elements the higher the enrichment of their isotopic compositions by heavy isotopes relative to SW composition. This may indicate that the processes resulting in the formation of P1 and P7 noble gas components are similar. Since ions in SW have the same velocity, the heavier the mass the higher the ion energy. During implantation, this results in different implantation ranges for elements and isotopes of different masses and thus produces element and isotope fractionation in solids where the ions are implanted, i.e., Q phase grains, if their grain sizes are comparable with implantation ranges[30]. This happens because a larger fraction of heavy ions go through without stopping inside the grains compared to that of the lighter ions. However, the direction of fractionation in this case (depletion of heavy gases and isotopes relative to the SW composition) is opposite to what is observed in P1 and P7. Thus, P1 and P7 do not represent fractionated SW components as a result of their direct implantation into Q grains.

Correlation between the difference in the elemental composition of P1 noble gases from that in SW with FIP (first ionisation potential)[3] indicates that ionisation is an important factor during the formation of the component. This is also true for P7. Among a number of different laboratory experiments aimed at reproducing the capture of noble gases in Q, the most promising are those in which ionised noble gases are trapped at low energy in a continuously growing surface[31] or organic compounds[32]. To a certain extent, these experiments can reproduce the observed element and isotope fractionation in P1. These experiments are related to the processes in the protoplanetary disk with solar rather than SW starting composition, i.e., the composition of the solar photosphere. During the formation of SW in the solar atmosphere it is mass fractionated relative to the composition in the outer convection zone and photosphere. According to the Inefficient

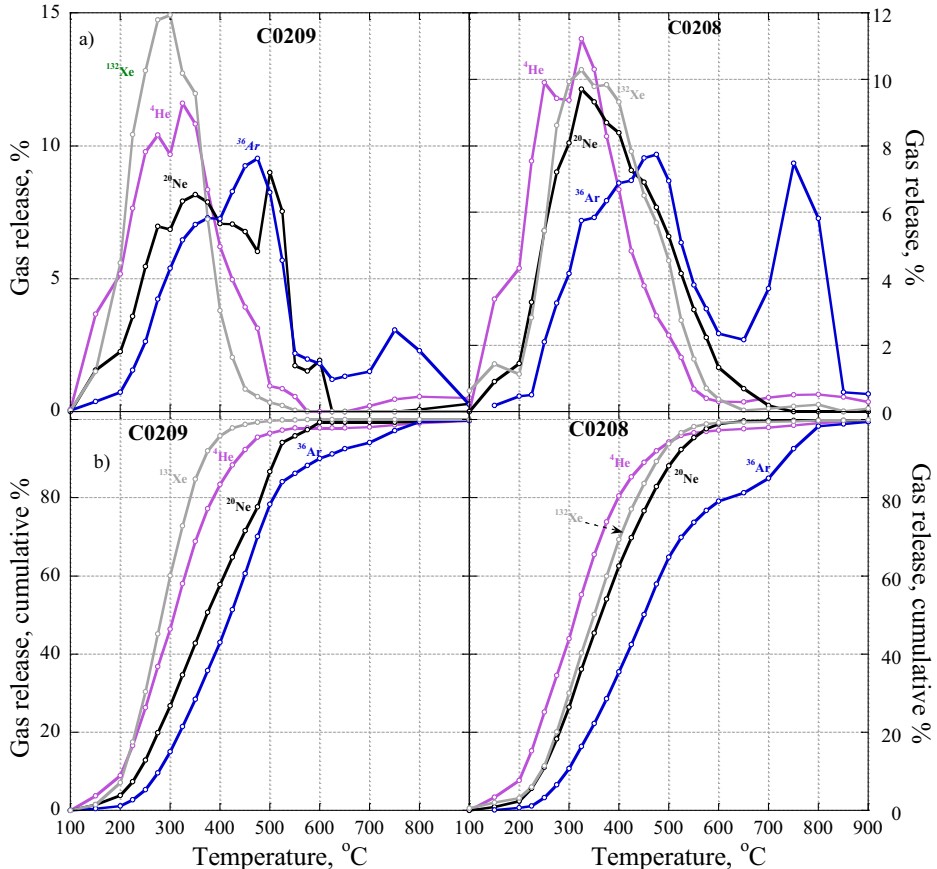

**Fig. 5 | Noble gas release patterns during stepped combustion of the C0209 and C0208 samples (Source data are provided in Supplementary Tables S1– S4).** **a** the upper panel shows differential releases while (**b**) – the low panel shows cumulative releases. The complex character of the release curves for He, Ne and Ar in C0209 is due to the fact that they are mixtures of several components, P1, SW and P7, in which P1 and SW predominate over P7. For Xe, the opposite is true: P7 is significantly dominate over other components. Xe is released at a slightly lower temperature from C0209 than from C0208 and ahead of He and Ne for the samples correspondingly.

Coulomb drag (ICD) model and calculations based on the correlation between fast and slow SW isotopic compositions with He/H ratio[33], Ne becomes slightly isotopically lighter in SW compared to solar photosphere (13.9 vs. 13.4 for $^{20}Ne/^{22}Ne$ ratio). For Xe, the solar photosphere isotopic composition was also estimated using the ICD model[3]. Compared to the photosphere, the elemental ratios in SW are fractionated in the opposite to the isotope fractionation direction since element ratios depend on the element's FIPs, which are higher for the lighter noble gases when compared to heavier ones[34], while isotope fractionation depends on ion masses. The range for solar composition is shown in Fig. 8. Thus, if carriers of P1 and P7 (Q phase) have been formed in the protoplanetary disk or protosolar molecular cloud, element and isotope fractionation during their formation should follow the directions as shown with arrows in Fig. 8, i.e., P1 is slightly or not at all fractionated in isotopic composition and strongly by elemental composition, while P7 is strongly fractionated by both isotopic and elemental compositions relative to solar composition.

The following questions remain, however: a) Why does P7 have a different degree of fractionation compared to P1? b) What does the difference in the degree of fractionation for P1 and P7 mean in terms of the fractionation mechanism and formation of their carrier(s)? As discussed above, Xe-P7 is very rare and therefore does not contribute significantly to the Xe budget in meteorites but provides clues to the origin of Xe-P1, since the formation of both involves mass fractionation processes of varying degrees. To reveal the clues, the above questions need to be answered. However, first, the presence of P7 has to be confirmed in other Ryugu samples (material from the Bennu asteroid,

which will soon be available for laboratory studies, is another potential source of the component) and includes all Xe isotopes. After that, in order to understand the chemical nature of the P7 carrier, it will be important to establish its reaction with acids or other solvents.

## Methods

### Finesse mass spectrometry system

Bulk samples were analysed using a Finesse mass spectrometric complex designed at the Open University[35,36]. The main feature of this instrument is the possibility to analyse simultaneously the isotopic compositions and contents of several elements (He, Ne, Ar, Xe, N, and C) extracted from a single sample. It consists of two magnetic sector mass spectrometers with a radius of 12 cm and one quadrupole mass spectrometer, all connected to a common extraction system. All mass spectrometers are operated in a static vacuum mode. One of the magnetic sector mass spectrometers is used for the analysis of carbon in the $CO_2$ form; the other for molecular N and Ar; and the quadrupole spectrometer, for He, Ne, and Xe. Gases are released by heating the sample in a double-vacuum furnace with an inner sample quartz glass tube, an outer corundum tube (the space between which is pumped out to forevacuum) and a silicon carbide heater allowing heating up to 1450 °C. Pure oxygen generated by heating CuO to 850 °C is used for oxidation, and the unused portion of the oxygen is resorbed by copper oxide at 450 °C. The oxygen pressure during oxidation is 5–10 mbar, and the combustion time is 0.5 h.

The released gases are separated into the He + Ne, $CO_2$ + Xe, and Ar + $N_2$ fractions using cryogenic traps (a glass finger for sorption of

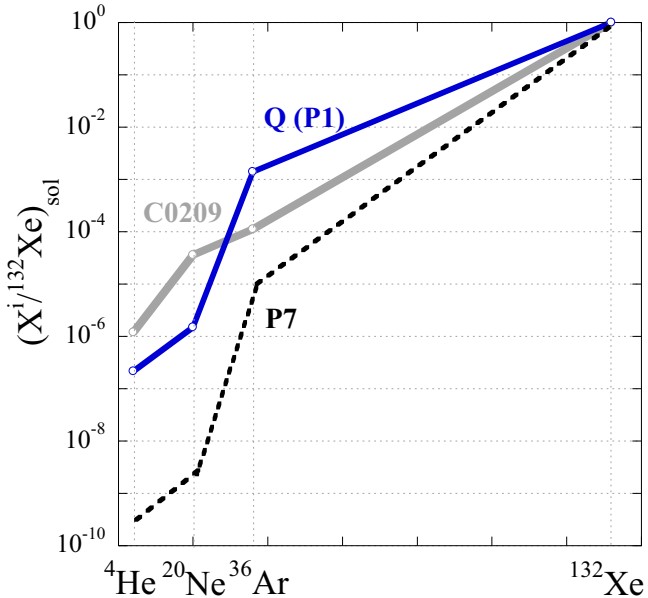

**Fig. 6 | Elemental composition of P1 and measured in C0209 (Source data are provided in Table 1) normalised to $^{132}$Xe and SW composition.** Estimated elemental composition of P7 is shown by a dotted line. This estimate is somewhat arbitrary but gives an idea of the possible elemental composition of this component (for details, see text).

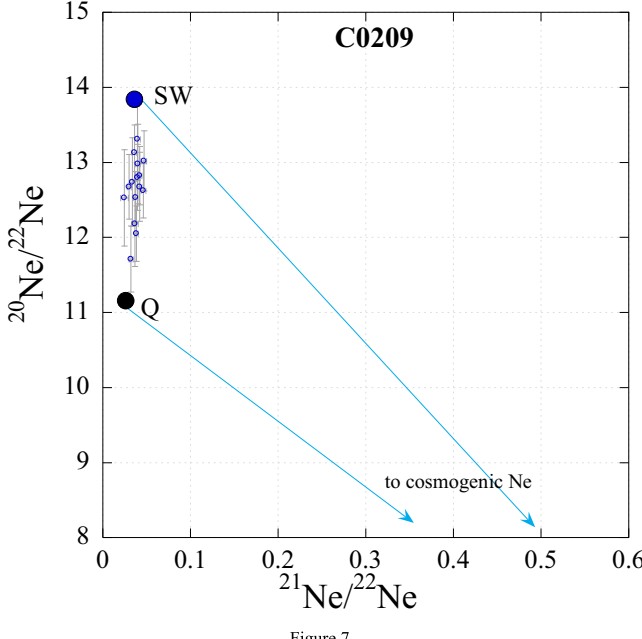

Figure 7

**Fig. 7 | Ne three-isotope diagram.** The open cercles with error bar are combustion steps for C0209 (Source data are provided in Supplementary Tables S1–S3). SW and Q compositions are also shown for reference. Ne isotope variations in the temperature steps are mostly due to mixing between SW and Q compositions with a small contribution of cosmogenic component. Alternatively, the variations can be associated with mass fractionation of implanted SW Ne during extraction from the sample due to different implantation ranges of Ne isotopes with different masses.

$CO_2 + Xe$, a cold finger with molecular sieves for sorption of $N_2$ and Ar, while He and Ne remain in the gas phase) and additionally purified on Al–Ti getters (for noble gases) and additional copper oxide (for nitrogen). Each of the $CO_2 + Xe$ and $Ar + N_2$ fractions is then subdivided into two approximately equal portions and used for the analysis of individual elements after purification. All the volumes of the vacuum system were calibrated, so the proportion of the total amount of gas included in one or another fraction is precisely known. The amount of the released $CO_2$ is determined by pressure measurements using a capacitance manometer Baratron™ with a precision better than 1%. Before introduction into the mass spectrometer, $CO_2$ is, if necessary, additionally split up to avoid exceeding the maximum permissible signal of the mass spectrometer amplifier. This is achieved by the calibration of the volumes in which gases are split and the determination of the mass spectrometer sensitivity. A similar procedure of nitrogen splitting is also applied and is based on the preliminary measurement of its amount in a small (5%) fraction using the quadrupole mass spectrometer in which sensitivity is also calibrated with reference gas. The mass spectrometers for $N_2$ and $CO_2$ are equipped with three collectors set for masses of 28, 29, and 30 and 44, 45, and 46, respectively. The measurement itself takes approximately one minute, during which ~300 data points were collected for each isotope that eventually provides a precision of 0.3–0.5‰. During measurements in a static vacuum mode, $CO_2$ is rapidly (half-life of ~20 s) transformed to CO; therefore, longer measurements are impracticable. The nitrogen signal also decreases during mass spectrometric measurements, although much slower than that of $CO_2$. For the calculation of $\delta^{13}C$ and $\delta^{15}N$, appropriate standards were measured alternately with the samples. To provide identical conditions for both measurements, the amount of the standard must correspond to that of the sample. For this purpose, a sampling system in which standard gas is continuously bleeding from a reservoir through a capillary is used. The pressure in the reservoir is about 3 bars to provide viscous flow. The necessary amount of gas is collected by setting the time of its

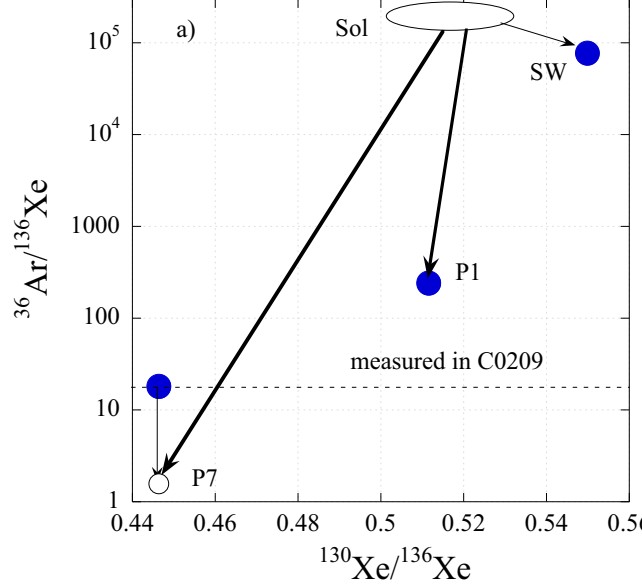

**Fig. 8 | Relationship between isotopic and elemental compositions for P7.** Compositions for other noble gas components (filled points) are shown for comparison. The open circle shows the estimated composition of P7 after correction for $^{36}$Ar from P1 (see text). SW composition is from Meshik et al., 2014[11]; P1 – from Busemann et al., 2000[4]. The area for solar (photosphere) composition is from Lodders et al., 2019[34] and Ott, 2014[3] (see text). The arrows show how the solar composition was fractionated to form SW, P1 and P7 components, as well as the direction for the true P7 composition. Source data for P7 are provided in Table 1 and Supplementary Table S5.

accumulation in a certain volume. This technique provides highly accurate (< 2%) correspondence between the amounts of sample and standard gases during measurements. Since isotope fractionation may occur in gas flow through the capillary, the standards are calibrated using either NBS standards (calcite for $CO_2$) or atmospheric nitrogen (for $N_2$) taken from fixed a volume gas pipette system. In addition, the gas flow rate is determined for all sampling systems of standards for determination of the sensitivity of mass spectrometers and the amounts of gas (nitrogen and noble gases) using the peak height method. The sampling system for noble gas standards (air) is similar and also calibrated in an appropriate manner.

The system described above is fully automated. For sample measurements, temperature and time of oxidation at each temperature step, the number of steps, and the list of elements (e.g., C, N, and Ar) are set. A few mg sample is loaded into a Pt foil (25 µm thick) capsule. The sample is dropped into an extraction furnace through a gate valve and heated at 100–200 °C for at least 0.5 h (the first step of oxidation was always conducted at the same100–200 °C to further decrease atmospheric contamination), and the analysis is further conducted in automated mode. Pneumatically controlled valves are used in the vacuum system, and cryotraps are cooled by pumping vapour of liquid nitrogen through a small cylinder jacket placed directly adjacent to a cold finger, which has good thermal contact with it. Cryotraps are equipped with a wire heater wound directly around the cold finger between it and the cooling jacket. The duration of the autonomous operation of the instrument during sample analysis depends on the volume of the liquid nitrogen Dewar. A 50-L Dewar is sufficient for continuous operation over 24 hours. During this time, 10–15 separate temperature fractions can be analysed depending on the set of elements analysed. The system blank was determined by the analysis of empty Pt foil. Since amounts of gases in the blank depending on the temperature, the blank experiments usually include a number of steps in the temperature range as for samples with, however, a lower temperature resolution.

### Sample preparation and loading into the Finesse system
The Ryugu samples we analysed are fine-grained aggregates, as shown in Supplementary Fig. S2. They were loaded into the vacuum system of Finesse without contact with the atmosphere. For that, we used a glove box filled with pure nitrogen. The microbalance for the determination of sample weight was accommodated in the glove box. The sealed containers with the Hayabusa2 samples were opened in the glove box. After putting the sample aliquot into a Pt capsule whose weight was determined before, the capsule was sealed (not vacuum-tight) and weighed again. Then the sample in Pt foil was transferred into a manifold via a portable gate valve again within the glove box. After closing the gate valve, the manifold was transferred to the Finesse lab and connected to the Finesse vacuum system so that the sample could be dropped into the extraction furnace through the gate valve after pumping.

### Xe isotope analysis on quadrupole mass spectrometer
For Xe isotope measurements, we used a Hiden Analytical quadrupole mass spectrometer (HAL 8 RC PIC-RGA 301#14168). The measurements were performed in static vacuum mode with a Ti-Al getter connected to the mass spectrometer chamber to maintain the static mode. The mass spectrometer mass resolution was set to completely resolve Xe masses without any tails on the neighbouring masses (Supplementary Fig. S3). At this resolution, Xe masses are not resolved from hydrocarbons. However, the contribution of hydrocarbons is practically insignificant even at Xe signals that are two orders of magnitude smaller than in the case of the analysis of the C0209 sample, which can be judged from the signals at masses 135 and 137, where the Xe signal is absent.

Peak jumping and ion counting were used to collect the data during 400 scans with 300 ms integration time and 100 ms between measurements at each peak. This took about 15 min to complete the measurement. During this time, Xe signal slightly decreases, but by no more than 10% (Supplementary Fig. S4). We measure only the 6 most abundant isotopes: from $^{129}Xe$ to $^{136}Xe$. At a signal rate of ~$10^4$ cps, as was observed during the C0209 sample analysis, precision at permil level can be achieved for Xe isotope ratios. Mass discrimination of the mass spectrometer for Xe is about 6.4‰/AMU with depletion of heavy isotopes (Supplementary Fig. S5), it remains stable at constant mass spectrometer settings and does not depend on the signal intensity within more than two orders of magnitude. We check this by analysing variable amounts of air Xe in the range comparable with that measured during the run of the C0209 sample (Supplementary Fig. S6). In other words, the variations in the isotope ratios during measurements of the sample with very high Xe concentrations cannot be explained by variations in the mass discrimination on the amount of Xe in the mass spectrometer.

### Reporting summary
Further information on research design is available in the Nature Portfolio Reporting Summary linked to this article.

## Data availability
Source data are provided in this paper.

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

## Acknowledgements

We are grateful to Jamie Gilmour and the anonymous reviewer for their constructive and useful comments that helped to improve the manuscript. We thank all the scientists and engineers of the Hayabusa2 project whose dedication and skills brought these precious particles back to Earth. We thank the UK Embassy in Tokyo for providing support for the transportation of Ryugu particles from Japan to the United Kingdom. M.I. acknowledges funding support from JSPS KAKENHI (grant numbers JP18K18795, JP18H04468, JP23H01238 and JPJSBP120235705). The Open University team acknowledges funding from STFC (grant #ST/X001180/1 and #ST/T000228/1) and the Royal Society (grant #IEC\NSFC\223102).

## Author contributions

A.B.V. analysed the samples, interpreted the obtained results and wrote the manuscript draft. F.A.J.A., I.A.F., Mahesh A., S.J.B., M.M.G., R.C.G., M.I. and M.S. contributed to the results interpretation and manuscript editing. R.C.G. also handled the samples in the glove box. N.T., M.U., A.Y., M.K., N.I., N.S., Takuji O., M.-C.L., Y.K., A.N., K.Y., K.U. and H.Y conducted the sample handling, preparation and mounting processes of Ryugu particles. A.M., M.N., M.Y., T.Y., Masanao A., T.U., S.-i.W. and Y.T. led JAXA curation activities for the initial characterisation of allocated Ryugu particles. S.N., Tatsuaki O., T. S., S.T., F.T. and M.Y. contributed to spacecraft operations and selection sample sides.

## Competing interests

The authors declare no competing interest.

## Additional information

## Consortium Phase2 curation team Kochi

Motoo Ito [2,3], Naotaka Tomioka [2], Masayuki Uesugi [4], Kentaro Uesugi [4], Akira Yamaguchi [5], Naoya Imae[5], Takuji Ohigashi [7,8], Yuzuru Karouji [11], Naoki Shirai[6], Hayato Yuzawa [7], Kaoki Hirahara[11], Ikuya Sakurai[14], Ikuo Okado[14], Ming-Chang Liu[9], Richard C. Greenwood [1], Ross Findlay[1], James A. Malley[1], Ian A. Franchi [1], Monica M. Grady [1], Alexander B. Verchovsky [1] ✉, Feargus A. J. Abernethy [1], Xuchao Zhao[1], Martin Suttle [1], Cerdic Pilorget[15], Jean-Pierre Bebring[15], Dambien Loizeau[15], John Carter[15], Lucie Riu[15], Tania Le Pivert-Jolivet[15], Katlyn McCain[16] & Nozomi Matsuda[16]

[14]Nagoya University, Synchrotron Radiation Center, Nagoya, Japan. [15]Institut d'Astrophysique Spatiale Centre Universitaire d'Orsay, Orsay, France. [16]University of California, Loss Angeles, USA.

