## [Peer Review File · Nature Communications]

REVIEWER COMMENTS

Reviewer #1 (Remarks to the Author):

This study by Verchovsky et al. reports noble gas data measured in three fine-grained asteroid grains from Ryugu brought back to Earth by the Hayabusa2 mission. A major discovery of this study is that the authors identified a new noble gas component (Xe-P7) present in one grain (C0209). This component has a distinct Ar/Xe elemental ratio than Xe-P1 the dominant noble gas component found in carbonaceous chondrites. Importantly the isotopic composition of xenon of this component is also distinct from Xe-P1 with a much more pronounced isotopic fractionation relative to solar xenon. This allows the authors to discuss the origin of noble gases in chondritic bodies which is an important goal in cosmochemistry since noble gases are excellent tracers of the origin and distribution of volatile elements in the nascent solar system and on terrestrial planets. The manuscript is overall well written and arguments justifying the existence of this new component are sound. Some figures (especially in the Supplementary material) would benefit from being redrawn and largely improved.

A major concern relies on the fact that the authors do not provide their data in the manuscript nor in the Supplementary Material (!). There should be a table with all data for temperature steps so that an interested reader can go look at them and also use them. Here we are only provided with bulk data despite the fact that most figures show data for temperature steps. I strongly recommend the authors to provide a clear table (.xls file with their submission) data and associated error bars for all temperature steps, all samples and of course all isotopic ratio determined in their study.

Another important concern relates to contamination by the terrestrial atmosphere. The authors state L91-92 that the samples "entirely avoided" exposure to air. This is too optimistic since the sample capsule briefly reopened during Earth's atmosphere entry leading to an almost air-like composition of sampled gas measured by Okazaki et al. (2022), 10.1126/sciadv.abo7239. Since air contamination cannot be excluded. What about fractionated air present in C0209 sample? Xenon is known to be sticky and adsorbs easily on fresh surfaces. If C0209 had highly reactive surfaces could the elemental and isotopic composition be explained by adsorption of fractionated Air-Xe?

I think this is the only point not fully addressed by the authors. They should show plots with fractionated air (taking some mass-dependent fractionation curve for absorbed Air Xe) and show that their data points advocated to reflect the presence of a Xe-X or Xe-P7 new components are not going through such a curve. This would demonstrate that Xe-P7 is not simply adsorbed fractionated air.

One major question arising from this study is why has this component been detected only in one Hayabusa2 grain? I acknowledge that only future studies will reveal if this component is present in other samples and this manuscript should ultimately be published to motivate the community.

The authors should also provide a plot similar to the one in Fig. 2 but for sample C0208 so that one can truly judge how different the two samples are in terms of isotopic ratios.

L57-58: maybe cite here the work by Heber et al. 2012 ApJ which discusses isotopic fractionation of solar gas ejected as SW.

L63: remove "apparently"

L71-73: here I think you should mention that the ^{129}Xe excess from the decay of ^{129}I is highly variable depending on the meteorite you take. It seems to correlate with C content suggesting a dilution effect of radiogenic ^{129}Xe by primordial i.e. less radiogenic xenon (see the recent study by Avive et al., 2022, *Geochemical Perspective Letters*, 10.7185/geochemlet.2228. They apparently found very primitive ^{129}Xe -poor xenon in C-rich meteorites such as Tarda).

L91-92: here you should probably modify "entirely avoided" exposure to air since the sample capsule briefly reopened during Earth's atmosphere entry leading to an almost air-like composition of gas measured by Okazaki et al. (2022), 10.1126/sciadv.abo7239.

legend Figure 5: it seems rather than Xe is released at lower temp for C0209 than for C0208

L213-214: and light isotopes should be measured to see if they confirm that Xe-P7 is a more fractionated Xe component than P1.

L243: ref. 27, likely wrong reference here. please correct.

L259-263: maybe also state here that a full determination of the isotopic composition of Xe-P7 is required (and will help to assess varying degrees of mass fractionation).

Fig. S2 looks very bad. Suggest taking a proper screenshot or even replotting the data. same for Fig. S3, this looks very bad, suggest taking raw data and making a real plot. also state in the legend if what we see is an air standard or a sample?

Reviewer #2 (Remarks to the Author):

(I also include this as a PDF which is formatted - I bolded the points I think are most important.)

Comments on "A new primordial noble gas component in the solar system discovered in the Ryugu asteroid" by Verchovsky et al.

The manuscript presents evidence for a new component of xenon identified in a sample of the asteroid Ryugu returned by the Hayabusa 2 mission. The existence of such a component would provide a new constraint on the mechanism by which noble gases got incorporated into the solid phase. Since these gases are used as tracers of the processes that lead from the solar nebula to planetary volatile reservoirs, such as atmospheres, it is an important observation and of interest beyond the noble gas community. The authors present a good case for the existence of the new component, convincingly ruling out terrestrial contamination. Although only 6 of the nine isotopes were reported, the high

concentration and the extent of mass fractionation are sufficiently demonstrated to warrant publication. The consistent ratios observed over a range of temperature steps suggest to me that it has been well captured by these analyses.

For this reason I think the paper deserves to be published, but I suggest some significant changes are needed.

It is vital that a paper that present a new component provide sufficient access to the source data for other researchers to study the evidence in detail. Unless I missed it (apologies if so) there is no data access statement. At minimum, the blank- and discrimination-corrected ratios and gas contents for each of the temperature steps for each of the samples discussed should be available. In addition, I suggest that the reader should be able to follow through the entire process from acquisition of raw data to generation of the finalized data, possibly using information preserved in a repository.

Also relating to the methods, the authors might clarify why ^{128}Xe is shown in Fig S2 but not reported; as I write I can't find information about the sample masses beyond the statement that they were 2-5 mg – please add them to Table 1; errors are described as standard error on the mean, so I assume that all error bars are 1σ - this should be stated somewhere (perhaps I missed it).

The authors identify the component because it is distinguished by an elevated xenon concentration and by the extent of mass fractionation of the xenon isotopes. According to Table 1 (which could be better formatted), the He concentration is within the range of reports by Okazaki et al, the Ne concentration is somewhat higher and the Ne has a higher $^{21}/^{22}$ ratio, the Ar content is within the range, but the Xe concentration is a factor of 10 higher; hence the Xe/Ar ratio is also a factor of 10 higher). In this it is similar to the Winchcombe analyses reported (I think) from the same lab in King et al Science Advances DOI10.1126/sciadv.abq3925. Supplementary Table 21 of that publication lists the Xe concentration as 6.7×10^{-7} cc/g, which is higher than that presented here, though no isotope data are reported. Could it be that this is the second time such a high concentration component has been detected? I think this should be noted and discussed.

The authors argue on the basis of mixing diagrams (Fig 2) that this is not a mixture of known components, and the panel dealing with ^{131}Xe suggests this is correct. ^{129}Xe is affected by decay of ^{129}I and so interpretation is complicated (see below), but the data do support the argument for a new component rather than a mixture. Later the authors note that the component appears to be related to solar xenon or Xe-P1 by mass fractionation, so it would be helpful if the general trend (gradient) of mass fractionation were included on each plot.

Turning to the implications, this new component will definitely present a puzzle to people working in this field: how does it relate to other xenon components and what does it tell us about the formation of the solar system and the source of the Earth's volatiles? It's not reasonable to expect the first paper to completely resolve all of these issues, but I think there are some areas where the manuscript's discussion needs to be strengthened.

The authors present a delta plot illustrating the different extents of mass fractionation exhibited by Xe-P7 and Xe-P1. Fig 4b illustrates an unusual feature of this component – it appears to have the same proportional excess of ^{129}Xe from ^{129}I decay as Xe-P1, even though xenon is present in a much higher concentration. This is puzzling. The authors argue that Xe-P1 and Xe-P7 are isotopically distinct, have different extents of elemental fractionation among the noble gases, and are present in carriers with different thermal properties, so why do they have the same proportionate enrichments in ^{129}Xe from ^{129}I decay relative to solar xenon? The authors need to discuss the implications of similar $^{129}\text{I}/^{132}\text{Xe}$ ratios (or (Excess $^{129}\text{Xe})/^{132}\text{Xe}$) ratios.

In addition, related to Fig 4b, I think some sort of statistical assessment as to whether Xe-P7 is consistent within error with mass fractionated Xe-P1 is needed – I don't find the figure very compelling that Xe-P7 is not simply mass fractionated Xe-P1 (this would still be interesting, but it might help explain the iodine result). For instance, in the past I used a simple calculation of chi-squared of the best fit mass fractionation (ref 9). This is not perfect since the errors are not independent (all the ratios include an error on ^{132}Xe) and so the chi-squared is lowered, but it at least it provides an indication. What is the corresponding minimum value of chi-squared for production of Xe-P7 by mass fractionation of Xe-P1? Does it inevitably drive us to conclude that this is not mass fractionated Xe-P1? There is some variability within Xe-P1 that is not captured by the composition reported in reference 4 (compare Fig 4 of reference 9) – against this variation is Xe-P7 significantly different from mass fractionated Xe-P1?

Around line 205 the authors discuss why Xe-P7 has not been observed before and propose that this is down to sample heterogeneity. I think this is unsatisfying – the samples are described as “fine grained” and are stated to be 2-5 mg (around 20-50 times the masses of the pellets previously analysed by the Hayabusa team). The thermal release patterns (Fig 5) do not suggest to me that thermal processing can efficiently remove Xe-P7 and leave Xe-P1 behind – given the high concentrations of Xe-P7 in its carrier (the measured concentration is a lower limit) would we not expect to have seen some trace of it in previous step heating studies? I do not have an answer to why this component is present in only this one sample but is it plausible that this component is found in this one aliquot to the extent that it dominates the Xe content, but not present at all in any of the others? Some sort of statistical discussion seems appropriate.

One final point. Generally, non-specialists (all but a few 10s of people at most) find extraterrestrial xenon isotope systematics confusing. There are a large number of components with esoteric names. The authors do a good job at focusing on the ones relevant to this work, but it might be useful to note that Xe-P1 is also referred to as Q-Xe or Xe-Q in the literature (e.g. the Busemann reference). I understand why the authors start calling it Xe-X then shift to Xe-P7, but I recall that Xe-X was used previously for (I think) what is now usually called Xe-HL, and I think a more direct ID as Xe-P7 from the start might be helpful (especially in Figs 3 and 4).

I would have found a table with the proposed Xe-P7 composition helpful – it was a while before I figured out it was listed in Fig 3.

Reviewer #1 (Remarks to the Author):

This study by Verchovsky et al. reports noble gas data measured in three fine-grained asteroid grains from Ryugu brought back to Earth by the Hayabusa2 mission. A major discovery of this study is that the authors identified a new noble gas component (Xe-P7) present in one grain (C0209). This component has a distinct Ar/Xe elemental ratio than Xe-P1 the dominant noble gas component found in carbonaceous chondrites. Importantly the isotopic composition of xenon of this component is also distinct from Xe-P1 with a much more pronounced isotopic fractionation relative to solar xenon. This allows the authors to discuss the origin of noble gases in chondritic bodies which is an important goal in cosmochemistry since noble gases are excellent tracers of the origin and distribution of volatile elements in the nascent solar system and on terrestrial planets. The manuscript is overall well written and arguments justifying the existence of this new component are sound. Some figures (especially in the Supplementary material) would benefit from being redrawn and largely improved.

A major concern relies on the fact that the authors do not provide their data in the manuscript nor in the Supplementary Material (!). There should be a table with all data for temperature steps so that an interested reader can go look at them and also use them. Here we are only provided with bulk data despite the fact that most figures show data for temperature steps. I strongly recommend the authors to provide a clear table (.xls file with their submission) data and associated error bars for all temperature steps, all samples and of course all isotopic ratio determined in their study.

We added tables (S1 and S2) with the data for all temperature steps for the analysed samples to the Supplementary materials.

Another important concern relates to contamination by the terrestrial atmosphere. The authors state L91-92 that the samples "entirely avoided" exposure to air. This is too optimistic since the sample capsule briefly reopened during Earth's atmosphere entry leading to an almost air-like composition of sampled gas measured by Okazaki et al. (2022), 10.1126/sciadv.abo7239.

This is not exactly true. Okazaki et al., 2022 determined the total pressure and gas composition in the sealed container holding the samples 30 hours after the return capsule entered Earth's atmosphere. The pressure in the canister was 68 Pa and gas composition suggests that the gases come from both terrestrial atmosphere, penetrated inside the container through the metal seals probably due to shock occurred at the time of parachute deployment, and from the samples. This means that the pressure of atmospheric gases penetrated into the sample container is $>10^3$ times less than that if the container would be fully exposed to the terrestrial atmosphere (10^5 Pa). Another words, the container remained under vacuum before the samples were removed from it and stored under pure N_2 atmosphere.

We added a short discussion about it to the main text (L92-98)

Since air contamination cannot be excluded. What about fractionated air present in C0209 sample ? Xenon is known to be sticky and adsorbs easily on fresh surfaces. If C0209 had highly reactive surfaces could the elemental and isotopic composition be explained by adsorption of fractionated Air-Xe ?

In terms of surface reactivity, the Hayabusa-2 samples should not differ from each other, and high concentrations of Xe would be observed in all of them if contamination with terrestrial Xe had occurred, which obviously is not the case.

I think this is the only point not fully addressed by the authors. They should show plots with fractionated air (taking some mass-dependent fractionation curve for absorbed Air Xe) and show that their data points advocated to reflect the presence of a Xe-X or Xe-P7 new components are not going through such a curve. This would demonstrate that Xe-P7 is not simply adsorbed fractionated air.

The arguments presented above appear to be enough to exclude association of Xe-X with atmospheric contamination. However, we added mass-fractionation lines through atmospheric composition in Figure 2. The best plot showing that Xe-X is not due to mass fractionation of air Xe is a plot of $^{130}\text{Xe}/^{132}\text{Xe}$ vs $^{136}\text{Xe}/^{132}\text{Xe}$. For even greater clarity, the plot shown below is added to the Supplementary materials (Fig. S1).

One major question arising from this study is why has this component been detected only in one Hayabusa2 grain? I acknowledge that only future studies will reveal if this component is present in other samples and this manuscript should ultimately be published to motivate the community.

We completely agree with the statement.

The authors should also provide a plot similar to the one in Fig. 2 but for sample C0208 so that one can truly judge how different the two samples are in terms of isotopic ratios.

There is such a plot in Figure 3 ($^{129}\text{Xe}/^{132}\text{Xe}$ vs Xe cumulative release) for the sample C0208 where one can see the difference with the sample C0209. The other data where such comparison can be made are present now in the table (S2) added to the Supplementary materials.

L57-58: maybe cite here the work by Heber et al. 2012 ApJ which discusses isotopic fractionation of solar gas ejected as SW.

In more details this point is discussed in the "Discussion" section where this paper is cited, therefore, we prefer to refer here to this section rather than to the paper (L57).

L63: remove "apparently"

OK.

L71-73: here I think you should mention that the ^{129}Xe excess from the decay of ^{129}I is highly variable depending on the meteorite you take. It seems to correlate with C content suggesting a dilution effect of radiogenic ^{129}Xe by primordial i.e. less radiogenic xenon (see the recent study by Avive et al., 2022, *Geochemical Perspective Letters*, 10.7185/geochemlet.2228. They apparently found very primitive ^{129}Xe -poor xenon in C-rich meteorites such as Tarda).

The important point is that we consider here Xe-P1, the primordial component formed in the early solar system and trapped into specific carrier phase Q, but not the bulk Xe composition in meteorites discussed in the paper by Avive et al. the reviewer refers to. At the time of Xe trapping ^{129}I was alive and trapped into Q along with Xe-P1. After this ^{129}I decayed into ^{129}Xe that resulted in enhanced $^{129}\text{Xe}/^{130}\text{Xe}$ ratio. This ratio depends on the I/Xe ratio at the time of trapping and the difference in trapping efficiency between Xe and I and should be the same for different meteorites provided that no loss of Xe from Q in variable amounts for different parent bodies occurred before all ^{129}I was converted to ^{129}Xe . The latter condition appears to be satisfied and, therefore, Xe-P1 identified in different meteorites does not show significant variations in $^{129}\text{Xe}/^{130}\text{Xe}$ ratio. Situation with meteoritic bulk Xe is different. In this case, Xe is a mixture of different components trapped by different carriers along with radiogenic ^{129}I . The carriers are not as resistant to thermal metamorphism as Q, and Xe may be lost before ^{129}I decays completely, causing the $^{129}\text{Xe}/^{130}\text{Xe}$ ratio to vary depending on the initial I/Xe ratio, timing and extent of Xe loss. In this context almost the same excess of ^{129}Xe in Q and Xe-X (P7) relative to the corresponding fractionation lines means that before trapping both components had similar I/Xe ratio and I and Xe were trapped with

similar relative efficiency into corresponding carriers. This discussion is added to the main text (L205-231).

L91-92: here you should probably modify "entirely avoided" exposure to air since the sample capsule briefly reopened during Earth's atmosphere entry leading to an almost air-like composition of gas measured by Okazaki et al. (2022), 10.1126/sciadv.abo7239.

See above.

legend Figure 5: it seems rather than Xe is released at lower temp for C0209 than for C0208

Yes, you are right. This misprint is corrected.

L213-214: and light isotopes should be measured to see if they confirm that Xe-P7 is a more fractionated Xe component than P1.

We have added a sentence as per the comment (L258-259).

L243: ref. 27, likely wrong reference here. please correct.

Corrected.

L259-263: maybe also state here that a full determination of the isotopic composition of Xe-P7 is required (and will help to assess varying degrees of mass fractionation).

Added (L306).

Fig. S2 looks very bad. Suggest taking a proper screenshot or even replotting the data. same for Fig. S3, this looks very bad, suggest taking raw data and making a real plot. also state in the legend if what we see is an air standard or a sample?

The comments have been accepted and appropriate changes applied (Figs S3 and S4 in "Supplementary material").

Reviewer #2 (Remarks to the Author):

(I also include this as a PDF which is formatted - I bolded the points I think are most important.)

Comments on "A new primordial noble gas component in the solar system discovered in the Ryugu asteroid" by Verchovsky et al.

The manuscript presents evidence for a new component of xenon identified in a sample of the asteroid Ryugu returned by the Hayabusa 2 mission. The existence of such a component would provide a new constraint on the mechanism by which noble gases got incorporated into the solid phase. Since these gases are used as tracers of the processes that lead from the solar nebula to planetary volatile reservoirs, such as atmospheres, it is an important observation and of interest beyond the noble gas community.

The authors present a good case for the existence of the new component, convincingly ruling out terrestrial contamination. Although only 6 of the nine isotopes were reported, the high concentration and the extent of mass fractionation are sufficiently demonstrated to warrant publication. The consistent ratios observed over a range of temperature steps suggest to me that it has been well captured by these analyses. For this reason I think the paper deserves to be published, but I suggest some significant changes are needed.

It is vital that a paper that present a new component provide sufficient access to the source data for other researchers to study the evidence in detail. Unless I missed it (apologies if so) there is no data access statement. At minimum, the blank- and discrimination-corrected ratios and gas contents for each of the temperature steps for each of the samples discussed should be available. In addition, I suggest that the reader should be able to follow through the entire process from acquisition of raw data to generation of the finalized data, possibly using information preserved in a repository. Also relating to the methods, the authors might clarify why ^{128}Xe is shown in Fig S2 but not reported; as I write I can't find information about the sample masses beyond the statement that they were 2-5 mg – please add them to Table 1; errors are described as standard error on the mean, so I assume that all error bars are 1σ - this should be stated somewhere (perhaps I missed it).

Tables with the results for all samples and temperature steps are added to the "Supplementary materials" section including sample weights, errors etc. (Table s S1 and S2). Figure S3 represents mass spectrum of air reference Xe for the amount of Xe comparable to those analysed in the sample C0209 (The amount of Xe is now indicated in the Figure's caption.

If we had known in advance that C0209 contained such large amounts of Xe, we would have included all the light isotopes in the list of mass numbers for

measurements. At the Xe concentrations as observed in the sample C0208, light isotope analysis does not make much sense due to high experimental uncertainties.

The authors identify the component because it is distinguished by an elevated xenon concentration and by the extent of mass fractionation of the xenon isotopes. According to Table 1 (which could be better formatted), the He concentration is within the range of reports by Okazaki et al, the Ne concentration is somewhat higher, and the Ne has a higher 21/22 ratio, the Ar content is within the range, but the Xe concentration is a factor of 10 higher; hence the Xe/Ar ratio is also a factor of 10 higher). In this it is similar to the Winchcombe analyses reported (I think) from the same lab in King et al Science Advances DOI10.1126/sciadv.abq3925. Supplementary Table 21 of that publication lists the Xe concentration as 6.7×10^{-7} cc/g, which is higher than that presented here, though no isotope data are reported. Could it be that this is the second time such a high concentration component has been detected? I think this should be noted and discussed.

There is an error in King et al. Supplementary Table S21: the Xe concentration should be 10^{-8} cc/g, not 10^{-6} as shown in the table, i.e. Xe concentration should be 0.672×10^{-8} or 0.00672×10^{-6} . 10^{-6} only applies to He, Ne and Ar.

Table 1 has been formatted to make its contents easier to read.

The authors argue on the basis of mixing diagrams (Fig 2) that this is not a mixture of known components, and the panel dealing with ^{131}Xe suggests this is correct. ^{129}Xe is affected by decay of ^{129}I and so interpretation is complicated (see below), but the data do support the argument for a new component rather than a mixture. Later the authors note that the component appears to be related to solar xenon or Xe-P1 by mass fractionation, so it would be helpful if the general trend (gradient) of mass fractionation were included on each plot.

Mass fractionation lines through atmospheric compositions were included for each plot in Figure 2.

Turning to the implications, this new component will definitely present a puzzle to people working in this field: how does it relate to other xenon components and what does it tell us about the formation of the solar system and the source of the Earth's volatiles? It's not reasonable to expect the first paper to completely resolve all of these issues, but I think there are some areas where the manuscript's discussion needs to be strengthened.

The "Discussion" section is entirely devoted to the discussion of these issues. How far can we take this discussion after first identifying the component in one sample? There are many other questions that need to be answered first before moving on

to more fundamental issues. Some of the questions are listed at the end of the manuscript. It's also unclear which areas Jamie is exactly referring to.

The authors present a delta plot illustrating the different extents of mass fractionation exhibited by Xe-P7 and Xe-P1. Fig 4b illustrates an unusual feature of this component – it appears to have the same proportional excess of ^{129}Xe from ^{129}I decay as Xe-P1, even though xenon is present in a much higher concentration. This is puzzling. The authors argue that Xe-P1 and Xe-P7 are isotopically distinct, have different extents of elemental fractionation among the noble gases, and are present in carriers with different thermal properties, so why do they have the same proportionate enrichments in ^{129}Xe from ^{129}I decay relative to solar xenon? The authors need to discuss the implications of similar $^{129}\text{I}/^{132}\text{Xe}$ ratios (or $(\text{Excess } ^{129}\text{Xe})/^{132}\text{Xe}$) ratios.

A discussion about the origin and implication of excess of ^{129}Xe in P1 and P7 is added (L205-231). The bottom line is that the same $(\text{excess } ^{129}\text{Xe})/^{132}\text{Xe}$ ratio in P1 and P7 means that both components had apparently the same initial/parent/source composition, including the $^{129}\text{I}/^{132}\text{Xe}$ ratio, before trapping into their carriers and no loss of Xe from the carriers occurred before all ^{129}I decayed.

In addition, related to Fig 4b, I think some sort of statistical assessment as to whether Xe-P7 is consistent within error with mass fractionated Xe-P1 is needed – I don't find the figure very compelling that Xe-P7 is not simply mass fractionated Xe-P1 (this would still be interesting, but it might help explain the iodine result). For instance, in the past I used a simple calculation of chi-squared of the best fit mass fractionation (ref 9). This is not perfect since the errors are not independent (all the ratios include an error on ^{132}Xe) and so the chi-squared is lowered, but it at least it provides an indication. What is the corresponding minimum value of chi-squared for production of Xe-P7 by mass fractionation of Xe-P1? Does it inevitably drive us to conclude that this is not mass fractionated Xe-P1?

Xe-P7 cannot be produced from Xe-P1 as a result of mass fractionation. The deviations from mass fractionation line or lines, if we allow some variations in their slopes) are too large compared to the experimental uncertainties. This is clearly seen in the plots c and d added to Figure 4. There is no need to use chi-squared calculations to confirm that.

There is some variability within Xe-P1 that is not captured by the composition reported in reference 4 (compare Fig 4 of reference 9) – against this variation is Xe-P7 significantly different from mass fractionated Xe-P1?

True. Considering Xe-P7 as a member of Planetary Xe family, a much larger variability in isotopic composition compared to that observed before (Fig.4 in ref. 9) has to be postulated. The reason for that is the difference in the degree of mass fractionation. An important question that can be asked in this connection is

whether the planetary noble gas family contains more than two members with variable degrees of mass fractionation.

Around line 205 the authors discuss why Xe-P7 has not been observed before and propose that this is down to sample heterogeneity. I think this is unsatisfying – the samples are described as “fine grained” and are stated to be 2-5 mg (around 20-50 times the masses of the pellets previously analysed by the Hayabusa team).

If we knew the degree of Ryugu regolith heterogeneity with respect to the P7 carrier, we could tell at what sample size it should be excluded. Otherwise, all arguments for and against will remain speculative.

The thermal release patterns (Fig 5) do not suggest to me that thermal processing can efficiently remove Xe-P7 and leave Xe-P1 behind – given the high concentrations of Xe-P7 in its carrier (the measured concentration is a lower limit) would we not expect to have seen some trace of it in previous step heating studies?

Again, with such a limited knowledge of the properties of P7, it is very difficult to argue about how effectively it can be removed. Instead of causing Xe-P7 to be released from its carrier, thermal processing may act more efficiently on the carrier itself.

I do not have an answer to why this component is present in only this one sample but is it plausible that this component is found in this one aliquot to the extent that it dominates the Xe content, but not present at all in any of the others? Some sort of statistical discussion seems appropriate.

We also have no answer to that, as well as we do not know what kind of statistical reasoning can be applied.

One final point. Generally, non-specialists (all but a few 10s of people at most) find extraterrestrial xenon isotope systematics confusing. There are a large number of components with esoteric names. The authors do a good job at focusing on the ones relevant to this work, but it might be useful to note that Xe-P1 is also referred to as Q-Xe or Xe-Q in the literature (e.g. the Busemann reference).

We added this (L60 and 64).

I understand why the authors start calling it Xe-X then shift to Xe-P7, but I recall that Xe-X was used previously for (I think) what is now usually called Xe-HL, and I think a more direct ID as Xe-P7 from the start might be helpful (especially in Figs 3 and 4).

We explained what exactly we mean when using name “Xe-X” in this context (L165-166). It does not seem appropriate to us to call the component “Xe-P7” before any relation to Xe-P1 is confirmed/established.

I would have found a table with the proposed Xe-P7 composition helpful – it was a while before I figured out it was listed in Fig 3.

A separate table (S3) with Xe-P7 isotopic composition is added to the Supplementary material section.

REVIEWERS' COMMENTS

Reviewer #1 (Remarks to the Author):

The authors addressed the concerns raised during the previous round of review and I do not have additional comments. This is an important discovery and these results should be published without delay in high-profile journal such as Nature Communications.

Reviewer #2 (Remarks to the Author):

For answers to the bulleted questions I refer to my previous review.

I think the authors have done a good job of revising the manuscript, which is significantly improved. I have some comments and think one minor point needs to be addressed (the first point below), but this can be overseen by the editor with no further external review.

Regarding the Winchcombe analysis, I think this has to be corrected in the literature somewhere as/before the current work is published. As things stand, another group would interpret their data quite differently because they would be under the impression that some Winchcombe samples have huge concentrations of xenon, and readers of the current paper will have the same question I had. It's not a big deal to point out a typo in the online material for the other paper, but it is necessary. (They may have done this already, I haven't checked.)

I thank the authors for giving my points about the discussion section due attention. (The areas I was referring to were the ones I then listed in my previous review – I'm sorry that wasn't clear.)

L204. I would argue it's not the case that Xe-P1/Q-Xe has the same composition in meteorites of different class – see Fig 4 of Ref 9. The other meteorites exhibit scatter along a mixing line to Xe-HL from fractionated solar wind. I would say the underlying fractionated solar component is the same, but that's not "Q-Xe" or "Xe-P1".

Fig 4: The lines are best fits, presumably this was done by minimizing chi-squared (e.g. by least square fitting). I think it's good practice to say what the minimum is, given that it has been calculated. Or they could quote MSWD.

Comment around L205. The carrier is present in one Ryugu sample but not the others. To illustrate, this seems possible if there is one carrier per ~2mg on average, but not if there are on average 1000 carriers per ~2mg. Given the fine grain size presumably the one carrier grain must be small; this suggests that the concentration in the carrier itself is huge. In contrast, Xe-P1 is present in roughly constant concentration in Ryugu samples, suggesting a large number of carriers per mg, and a correspondingly lower (but still very high) concentration in the carrier itself. Unless there was something odd about the sample (e.g. it had a significant volume that was anomalous in some way). I guess a lesson for us all is that we need to

try to characterise these samples more before analysing them, perhaps the authors should emphasise this as a recommendation for future work.

Reviewer #2 (Remarks to the Author):

For answers to the bulleted questions I refer to my previous review.

I think the authors have done a good job of revising the manuscript, which is significantly improved. I have some comments and think one minor point needs to be addressed (the first point below), but this can be overseen by the editor with no further external review.

Regarding the Winchcombe analysis, I think this has to be corrected in the literature somewhere as/before the current work is published. As things stand, another group would interpret their data quite differently because they would be under the impression that some Winchcombe samples have huge concentrations of xenon, and readers of the current paper will have the same question I had. It's not a big deal to point out a typo in the online material for the other paper, but it is necessary. (They may have done this already, I haven't checked.)

I thank the authors for giving my points about the discussion section due attention. (The areas I was referring to were the ones I then listed in my previous review – I'm sorry that wasn't clear.)

L204. I would argue it's not the case that Xe-P1/Q-Xe has the same composition in meteorites of different class – see Fig 4 of Ref 9. The other meteorites exhibit scatter along a mixing line to Xe-HL from fractionated solar wind. I would say the underlying fractionated solar component is the same, but that's not "Q-Xe" or "Xe-P1".

We corrected this paragraph and discuss only $^{129}\text{Xe}/^{130}\text{Xe}$ ratio in relation to the underlying mass-fractionated composition (L203-L208).

Fig 4: The lines are best fits, presumably this was done by minimizing chi-squared (e.g. by least square fitting). I think it's good practice to say what the minimum is, given that it has been calculated. Or they could quote MSWD.

The calculated chi-squared values for the fitting lines were added to the Figure 4 caption (L597-611).

Comment around L205. The carrier is present in one Ryugu sample but not the others. To illustrate, this seems possible if there is one carrier per ~2mg on average, but not if there are on average 1000 carriers per ~2mg. Given the fine grain size presumably the one carrier grain must be small; this suggests that the concentration in the carrier itself is

huge. In contrast, Xe-P1 is present in roughly constant concentration in Ryugu samples, suggesting a large number of carriers per mg, and a correspondingly lower (but still very high) concentration in the carrier itself. Unless there was something odd about the sample (e.g. it had a significant volume that was anomalous in some way). I guess a lesson for us all is that we need to try to characterise these samples more before analysing them, perhaps the authors should emphasise this as a recommendation for future work.

In response to these comments, we slightly extended discussion on the subject (L276-L291).